# Utilizing a National Wastewater Monitoring Program to Address the U.S. Opioid Epidemic: A Focus on Metro Atlanta, Georgia

**DOI:** 10.3390/ijerph20075282

**Published:** 2023-03-28

**Authors:** Tamara Wright, Atin Adhikari

**Affiliations:** 1University College, University of Denver, 2211 South Josephine Street, Denver, CO 80208, USA; 2Jiann-Ping Hsu College of Public Health, Georgia Southern University, 501 Forest Drive, Statesboro, GA 30460, USA; aadhikari@georgiasouthern.edu

**Keywords:** wastewater-based epidemiology, public health, opioids, wastewater discharges

## Abstract

The opioid epidemic has continued to be an ongoing public health crisis within Metro Atlanta for the last three decades. However, estimating opioid use and exposure in a large population is almost impossible, and alternative methods are being explored, including wastewater-based epidemiology. Wastewater contains various contaminants that can be monitored to track pathogens, infectious diseases, viruses, opioids, and more. This commentary is focusing on two issues: use of opioid residue data in wastewater as an alternative method for opioid exposure assessment in the community, and the adoption of a streamlined approach that can be utilized by public health officials. Opioid metabolites travel through the sanitary sewer through urine, fecal matter, and improper disposal of opioids to local wastewater treatment plants. Public health officials and researchers within various entities have utilized numerous approaches to reduce the impacts associated with opioid use. National wastewater monitoring programs and wastewater-based epidemiology are approaches that have been utilized globally by researchers and public health officials to combat the opioid epidemic. Currently, public health officials and policy makers within Metro Atlanta are exploring different solutions to reduce opioid use and opioid-related deaths throughout the community. In this commentary, we are proposing a new innovative approach for monitoring opioid use and analyzing trends by utilizing wastewater-based epidemiologic methods, which may help public health officials worldwide manage the opioid epidemic in a large metro area in the future.

## 1. Introduction

In 2020, 92,000 people died from drug overdoses, making it a leading cause of injury-related death in the United States, with 75% of those deaths attributed to prescription drugs or illicit opioids. Drug overdose deaths continue to increase in the United States, and the cause of these deaths has developed into a major public health crisis [1]. As of today, wastewater treatment plants (WWTPs) are not required to monitor opioid constituents, so opioids are not treated and removed from the wastewater. Unfortunately, the treated wastewater, also known as effluent, is discharged into tributaries (i.e., creeks, lakes, and rivers) where marine life resides. Correspondingly, with the increase in opioids within the effluent, marine health is also suffering significantly [2]. 

The opioid epidemic in Metro Atlanta continues to cause thousands of overdoses and deaths throughout the community per annum. In 2018, the Georgia Bureau of Investigation (GBI) conducted an analysis that determined more people in the Metro Atlanta area abuse opioids than anywhere else in the state [3]. Furthermore, a public health analysis for the 10-county Atlanta region between 1999 and 2016, conducted by the Atlanta Regional Commission (ARC), determined that the rate of opioid-related overdose deaths in Metro Atlanta has nearly doubled since 2013 and more than tripled since 2006 [4]. The ten counties that compose Metro Atlanta are Cherokee County, Clayton County, Cobb County, Dekalb County, Douglas County, Fayette County, Fulton County, Gwinnett County, Henry County, and Rockdale County. From 1999 to 2016, the overall overdose death rate and opioid-related deaths have significantly increased each year throughout Metro Atlanta [4] (Figure 1). Correspondingly, within an analysis conducted by the Georgia Department of Public Health, it was determined that there was an increase in opioid-related deaths in teenagers with age ranges from 15 to 19 years old in 2021 [5]. Nonetheless, public health officials and policy makers within Metro Atlanta have continued to increase awareness, education, access to treatment, outbreak detection and utilize naloxone to address the ongoing opioid epidemic [6]; but with the opioid overdose-related deaths increasing each year, local officials have been unsuccessful in providing long-term solutions to address this public health crisis. 

Opioid metabolites can be located within the wastewater throughout Metro Atlanta to assess opioid use. WWTPs have the ability to monitor opioid metabolites. Requiring municipalities within Metro Atlanta to monitor and report opioid contaminants that are discharged within their WWTPs can assist local and state government efforts in fighting the opioid epidemic within this area.

Starting in 2013, the areas within Metro Atlanta that have been significantly impacted by opioid overdose deaths are the more affluent areas which include Fulton County, Cobb County, and Gwinnett County (see Figure 1). In 2010, the Georgia Department of Public Health’s Drug Surveillance Unit attributed the increase in opioid deaths to the use and misuse of prescription opioids, such as oxycodone and hydrocodone. Furthermore, in 2013, the use and misuse of illicit opioids (e.g., heroin and fentanyl) increased within the affluent areas of Metro Atlanta (e.g., Cobb County, Fulton County, and Gwinnett County), which caused a significant increase in opioid overdose deaths [7]. Considering the enormous impact of opioid overdose deaths, it is imperative that large metro areas, such as the one comprising Atlanta, establish a drug surveillance system that monitors the use and deaths related to opioids and other harmful substances. 

## 2. Public Health Surveillance Tools

### 2.1. Conventional Surveillance Methods That Are Utilized to Assess Opioid Use and Opioid-Related Deaths

Public health officials have utilized various conventional surveillance methods to assess and evaluate trends related to opioid use, illicit opioid use, and opioid-related deaths. Some of the most common conventional surveillance methods that are still being used today are self-reported surveys, clinical testing, medical records, prescription data, human biomonitoring studies, crime statistics, and risk assessment tools [8]. Within Georgia, some common interactive tools that are used by Georgia public health officials to assess opioid use and opioid deaths are the Georgia Prescription Drug Monitoring Program (PDMP) and the Georgia Strategic Prevention System (GASPS) [7]. In addition, there are various interactive surveillance tools that are used across the nation to provide an overview of community health in regard to opioid use and overdoses, which include the Online Analytical Statistical Information System (OASIS), the CDC’s Drug Overdose Surveillance and Epidemiology (DOSE) Data Dashboard, the CDC’s SUDORS Dashboard: Fatal Overdose Data, and the NEMSIS Opioid Overdose Surveillance Dashboard [7].

Historically, conventional surveillance methods have been effective with providing an overview of community health data. Unfortunately, people who abuse opioids rarely disclose their opioid use facts due to penalties and negative consequences associated with self-reporting substance abuse [9,10]. Withholding this information makes it difficult for public health officials to accurately collect data related to opioid use within a community. To increase the validity of opioid use data, wastewater-based epidemiology could be used as an additional surveillance tool.

### 2.2. Utilizing Wastewater-Based Epidemiology as a Public Health Surveillance Tool

Wastewater-based epidemiology (WBE) is a public health surveillance tool that has been proposed to present a snapshot of the overall health of a community based on what is being excreted in a pooled sample of sewage [8,11,12]. WBE is an approach that was introduced in 2001 to monitor illicit drugs [8,11], and it was first applied in 2005 in four WWTPs in Italy to monitor and track trends regarding cocaine use [13]. Biobot Analytics, a United States based company, was the first organization to introduce WBE into the market as a technique to solve and monitor public health issues (e.g., alcohol, tobacco, exposure to parabens, infectious diseases, and SARS-CoV-2) [14,15,16,17] within various communities, and their efforts have continued to have a positive impact around the world [18].

In comparison to conventional surveillance methods, WBE provides a more accurate snapshot of the health of the community through a pooled sample. When using conventional methods, data can take months to years to collect, and the health of the community may not be accurate [8]. On the contrary, WBE provides real-time data of the health of the population at any given time [8,10,11]. WBE methods are also less invasive, and users of opioids are not easily identifiable [8]. Moreover, WBE does not rely on self-reporting data or a database of prescribed opioids; which adds to the rationale that WBE is a necessary complementary tool that can coincide with conventional surveillance methods [8]. 

Utilizing WBE as a public health surveillance tool also has several limitations in comparison to conventional surveillance methods. Target indicators are not accurately identified, and the sources of disposal of opioids and usage patterns are unknown [8]. There are also several limitations that are associated with the analytical techniques. When using conventional surveillance methods, such as clinical testing, the data and analysis of opioids are more accurate compared to the wastewater analysis [8,19]. Wastewater analysis also has the potential to be impacted by environmental factors (e.g., degradation) or interferences within the matrix used within the analysis [8,19]. In addition, further research is necessary to determine the costs and benefits of utilizing WBE in comparison to conventional public health surveillance tools [8]. As the national opioid crisis continues to be detrimental to the health of the public, public health officials will continue to explore various solutions to provide awareness and to reduce the impact opioids have on all people across the United States. In comparison to conventional public health surveillance tools, WBE is a practical, effective application that can provide real-time data to assist public health officials on how to address the opioid epidemic within their communities.

## 3. Practical Applications and Benefits of Wastewater-Based Epidemiology

Recently, the U.S. Department of Health and Human Services established a national disease surveillance program, leveraging wastewater epidemiology by collaborating with Biobot Analytics. In 2021, Biobot Analytics collaborated with Ginkgo Bioworks, a United States based company, to analyze wastewater samples to track the spread of COVID-19 within all fifty states. Biobot Analytics has also assisted several researchers from various organizations and policy makers from various cities, with monitoring and tracking opioids within their wastewater [20,21,22]. Biobot Analytics has demonstrated that beyond COVID-19, wastewater-based epidemiology can be expanded to monitor seasonal influenza outbreaks, and proactively detect novel viral pathogens and antibiotic resistance [22]. One of Biobot Analytics’ initial projects utilizing WBE techniques was in 2017 within the Town of Cary, North Carolina. In 2015, there were more than 1100 opioid-related deaths, and in 2017, the Town of Cary saw a 40% increase in fatal overdoses and a 135% increase in non-fatal overdoses over the previous year [23]. Similar to Metro Atlanta, the Town of Cary has experienced a 73% jump in opioid-related deaths since 2005. In 2017, the Town of Cary in North Carolina was one of the first cities within the U.S. that conducted a pilot study that used wastewater monitoring to track opioid consumption. The results from this project determined that prescription opioids and fentanyl were the main opioids found in the wastewater. Furthermore, the results from this project increased awareness about the opioid crisis in their community, and reduced barriers and stigma for those who might want to seek support [23].

There is a plethora of research on the benefits of wastewater-based epidemiology [24,25,26]. Gushgari et al. discussed how wastewater-based epidemiology was used to assess drug consumption on a university campus [25]. Within the wastewater samples that were analyzed using liquid chromatography with tandem mass spectrometry (LC–MS/MS), indicators of consumption of morphine, codeine, oxycodone, heroin, fentanyl, methadone, buprenorphine, amphetamine, methylphenidate, alprazolam, cocaine, and MDMA were discovered. Gushgari et al. [25] determined that utilizing WBE is a successful tool that can be used within collegiate settings to provide near real-time data on opioid use throughout the campus. An article published by researchers from the Centers for Disease Control and Prevention (CDC) discussed how they utilized wastewater-based epidemiology to engage pharmacies in the ongoing opioid epidemic response [20]. The results from this study assisted the local community in targeting new policies and programs, facilitating partnerships between stakeholders involved in opioid response efforts, monitoring the effect of interventions on community health over time, and tailoring educational materials to the substances being consumed in each community [20]. Utilizing a combination of survey-based monitoring and wastewater-based epidemiology can assess opioid use and abuse more comprehensively within a community, and potentially support governments in developing policies to scale down opioid use [27].

## 4. Utilization of Global National Wastewater Monitoring Programs to Monitor Drug Use and Consumption

### 4.1. Global Wastewater Monitoring Policies on Opioid Use

Wastewater analysis has been utilized by various countries to assess trends in opioid use and to provide solutions that can address the worldwide opioid epidemic. Currently, there are a few countries that monitor opioid use through the use of their national wastewater monitoring system, but primarily, opioid use is monitored voluntarily through wastewater analyses conducted by researchers. Moreover, countries do not require WWTPs to report opioid metabolites that are found within their wastewater discharges. Considering the fact that countries do not require WWTPs to report opioid metabolites found within their influent and effluent discharges, the approach presented within this paper is a novel idea that can be streamlined globally.

Countries across the globe have adopted similar wastewater monitoring processes to assess opioid metabolites within the sewage throughout the various communities. Oftentimes, wastewater samples are collected within areas known for illicit drug use and abuse, and opioid overdoses or pilot studies will be conducted to determine if opioid use is a significant issue within a specific area. A widespread wastewater-based epidemiology sampling approach that is used globally is composite and grab sampling [28,29]. Composite sampling is the collection of an aliquot at a specific time each hour, over the course of a 24-h period. Researchers have used the grab sampling approach by manually collecting samples at manholes throughout their desired areas [20,23]. Due to the heterogeneity of the composite sampling approach, composite sampling is favored over grab sampling [30]. Even though these sampling methods have proven to be efficient, since opioid monitoring at WWTPs is not a widespread requirement, these sampling methods are conducted by researchers throughout the world on a voluntary basis.

### 4.2. Nationalized Wastewater Monitoring Programs to Assess Substance Use and Abuse

Numerous countries worldwide have used and established national wastewater monitoring systems to assess opioid use within affected areas throughout their country. Some of the countries that have yielded the most success with using WBE approaches through their national wastewater monitoring programs are within Europe, Canada, and Australia. In 2016, the Australian Criminal Intelligence Commission established the National Wastewater Drug Monitoring Program to monitor illicit and licit drug consumption and abuse. The main goal of this program is to collaborate with government, law enforcement, WWTPs, public health officials, and community organizations to analyze drug trends and implement new policies to combat this epidemic [31]. An annual report is generated each year that provides information on the 12 substance metabolites (alcohol, ketamine, nicotine, heroin, 3,4-methylenedioxy methylamphetamine (MDMA), 3,4-methylenedioxyamphetamine (MDA), cocaine, cannabis, oxycodone, fentanyl, amphetamine) that were discovered through wastewater analysis from the WWTPs. Within the 2022 report, the data was collected from 56 WWTPS, which equated to a coverage of 56% of the population (14.1 million Australians) [32]. Since the start of this program, Australia has seen an upward trend in the consumption of MDMA, fentanyl, heroin, and cocaine, and policy makers have been able to provide resources to these areas that are significantly impacted by substance consumption [32].

Correspondingly, the Public Health Agency of Canada (PHAC) has been in collaboration with departments, municipal governments, and academia throughout Canada to monitor SARS-CoV-2 through wastewater surveillance. The data discovered from the wastewater surveillance and analysis were generated and published in the Canadian Wastewater Survey (CWS). The CWS was initially formulated to detect the onset of COVID-19 throughout the various communities, with hopes of providing a rapid public health response to the community. Evaluation of the CWS data from the beginning of the COVID-19 pandemic determined that there had been a significant increase in drug consumption and opioid overdoses. The CWS collected wastewater samples from WWTPs located in Halifax, Montréal, Toronto, Edmonton, and Vancouver from March to July 2019 and from January to July 2020. The CWS determined that the levels of cannabis, fentanyl, and methamphetamine significantly increased since the start of the pandemic [33]. Since the publication of this report, the Government of Canada has established several policies and procedures to address the opioid crisis through the increase in awareness, prevention, enforcement, funding, and access to treatment [34].

Moreover, in 2016, the European Monitoring Centre for Drugs and Drug Addiction (EMCDDA) developed a report that discussed the comprehensive review and benefits of assessing illicit drugs through the use of wastewater-based epidemiology throughout Europe [30]. This report highlighted the significance and benefits of WBE applications worldwide, the importance of improving ethical challenges associated with WBE, and the need to discover a streamlined approach to improve the credibility and scalability of WBE studies that are conducted worldwide [35]. Furthermore, the EMCDDA emphasized the importance of a collaboration among chemists, environmental engineers, epidemiologists, pharmacologists, and addiction and intervention groups worldwide to monitor opioid use and provide enhanced evaluation interventions for the affected population [30].

Comparably, in 2020, the Centers for Disease Control and Prevention (CDC) established the National Wastewater Surveillance System (NWSS) to track the presence of SARS-CoV-2 to provide early detection and coordinate a rapid response to affected areas throughout the United States [36]. Currently, the United States has not utilized the NWSS or any other national wastewater monitoring programs, such as the National Pollutant Discharge Elimination System (NPDES) (NPDES}, to monitor opioid use and abuse because public health policy makers are probably unaware of this potential promising application. Other countries have demonstrated the benefits and importance of establishing national wastewater monitoring programs to assess trends and provide valuable insights regarding opioid use and abuse. Within this paper, we have proposed a national framework that can be implemented to monitor opioid use and abuse within affected communities throughout the United States.

## 5. Proposed Framework for Monitoring the Opioid Epidemic in Metropolitan Areas via the National Wastewater Monitoring Program in the U.S.

### 5.1. Current Process of Monitoring Wastewater Discharges within the United States

The National Pollutant Discharge Elimination System (NPDES) permit program was established by the Environmental Protection Agency (EPA) under the Clean Water Act (CWA). The NPDES permit program addresses water pollution by regulating point sources that discharge pollutants to waters of the United States [37]. The CWA authorizes the EPA to monitor and enforce permitting and administrative procedures for each state within the U.S.

In Georgia, the EPA has authorized the Georgia Department of Natural Resources (GADNR) the ability to issue and monitor NPDES permits throughout the state under the CWA. Within each permit, there are requirements for standard conditions (e.g., duty to reapply, duty to comply and provide information, operator certifications, and inspections, entry, and records), permit conditions (e.g., technology-based limitations, water quality-based limitations, and special conditions), and reporting conditions (e.g., electronic monitoring reports, annual reports, name changes and ownership transfers, and other non-compliance). The primary discharge pollutants that are monitored on the GA NPDES permits include pH, Total Suspended Solids (TSS), Biochemical Oxygen Demand (BOD), ammonia, total phosphorus, oil and grease, turbidity, dissolved oxygen, temperature, and fecal coliforms. Currently, wastewater treatment plants in Metro Atlanta are not required to report opioid metabolites to the GADNR as part of the requirements for the GA NPDES permit. The proposed solution for this public health crisis is increased monitoring of opioid use through wastewater surveillance by way of NPDES permits.

### 5.2. Integration of Opioid Discharge Monitoring within NPDES Permits

Wastewater-based epidemiology is an innovative, effective approach that can be utilized to monitor opioid use within the Metro Atlanta region to reduce the impacts on public health. To reiterate, the EPA established the NPDES permit program and grants authorized environmental protection divisions to enforce the program within each state. The NPDES permit program was established in congruence with the Clean Water Act to prevent point and non-point-source discharges from impacting local watersheds. In relation to the state of Georgia, the GADNR has been granted authority by the EPA to grant permits under the NPDES permit program to WWTPs throughout the state.

Currently, municipalities within Metro Atlanta are not required to monitor or report opioid contaminants that are discharged within their wastewater treatment plants. The proposed solution for this public health crisis is to increase monitoring of opioid use by using a combined approach of wastewater surveillance and NPDES permit requirements. This process would require WWTPs within Metro Atlanta to report the various opioid contaminants to the Georgia Environmental Protection Division as part of the NPDES regulations (see Figure 2).

As part of the proposed wastewater monitoring process, a collective effort will be critical among the WWTP employees (i.e., wastewater laboratory analysts, wastewater operators, and supervisors) and the municipal permitting manager at the GADNR. The wastewater operators will collect the required NPDES permit influent and effluent composite samples each week. The wastewater laboratory analysts will analyze the composite samples and report the results as required on the NPDES permit. The GA NPDES municipal permits are renewed and revised every five years, and as part of the renewal process, the renewed NPDES municipal permits within the Metro Atlanta area will contain opioid metabolites. Researchers have established a list of opioids that should be monitored within the WBE process [20,21,38], and these opioid discharges would be monitored on the NPDES permit. These opioids include codeine, morphine, heroin, hydrocodone, hydromorphone, oxycodone, oxymorphone, fentanyl, meperidine, methadone, buprenorphine, tramadol, acetaminophen, and naloxone. The monthly report will be submitted into the electronic data reporting system, also known as the Network Discharge Monitoring Report (NetDMR). The monthly NetDMR will provide data on the various opioid discharges that were analyzed each month. The information on the monthly NetDMR can be used by the local public health officials, scientists, and researchers to implement more strategies to combat the ongoing public health crisis. We believe this will be an innovative approach for the Metro Atlanta because the proposed solution has not been suggested or implemented within other states or municipalities throughout the United States (see Figure 2).

## 6. Evaluation of Monitoring Opioid Discharges

### 6.1. Impact of WBE Approach on Public Health and the Environment

The literature supports that opioid use has a significant adverse effect on public health, and that wastewater surveillance is an innovative public health tool that can be used to significantly reduce the impacts associated with this crisis [20,21,25]. Utilizing wastewater-based epidemiology as part of the NPDES permit program to monitor the emergence of opioid use and overdoses in metropolitan areas could significantly reduce the occurrence of this endemic. The research presented in this proposal has indicated that wastewater-based epidemiology is an effective and innovative solution that can be used to address the opioid epidemic within various communities. A combination of utilizing complementary public health data sources, opioid prescriptions, opioid misuse, and wastewater-based epidemiology can assist public health officials with identifying prevention measures, and allocating the appropriate resources [8,39]. Moreover, adding opioid metabolites to the GA NPDES permits will allow wastewater treatment plants to assist public health officials and policy makers with developing data on the areas that are most impacted by opioid use.

In addition, if opioid discharges are not monitored by the NPDES permit program, not only is marine health impacted, but these discharges could have an impact on water reuse processes and wastewater discharges that enter connecting waterways (e.g., lakes, streams, creeks, and rivers) that are used for recreational use, such as fishing and swimming. The GADNR has established requirements for water reuse but ensuring that opioid contaminants are not within the reuse water is not a requirement. Monitoring and treating these opioids that travel through the sewage will reduce the number of opioids that are ingested by the animals within surrounding aquatic ecosystems. Removing opioid metabolites that reside within the wastewater through chemical, biological, and physical processes, will also prevent any health effects that may result from human consumption of these exposed animals within the aquatic ecosystem. Likewise, without monitoring opioid discharges from the wastewater treatment plants, water quality could have long-term impacts on marine health and public health.

### 6.2. Feasibility and Social Issues Associated with Using WBE Techniques

The implementation of monitoring opioid discharges by using WBE is feasible to address the opioid epidemic and to prevent environmental impacts. The proposed solution cannot be executed effectively without a collaborative effort from the people in the community, representatives from the GADNR, employees at the WWTPs in Metro Atlanta, and public health officials. Regular communication among WBE practitioners, epidemiologists, and public health officials is needed to ensure that wastewater results monitoring opioid use inform broader policy responses and that practitioners adjust their public health approaches and responses to align with immediate needs that address the ongoing opioid epidemic [40]. WBE tools should be restricted to settings where performing swift, directed interventions that includes as much of the entire population-of-interest as possible [40].

There are also various social issues that are associated with utilizing wastewater-based epidemiology to address community health issues. The location of a wastewater sample (treatment facilities, pumping stations, manholes, and other sewer locations) can be representative and provide information on the disease burden of millions of residents within a community [41]. Even though a wastewater sample is a critical representation of opioid use within a community, it does not provide contextual information, such as demographics, pattern of use, and administered routes that can assist with intervention strategies [8,42]. Understanding the problem areas through wastewater sampling and analysis allows community leaders to understand the areas that require direct assistance and interventions. Utilizing wastewater-based epidemiology tools could help officials locate areas within counties that are impacted most prevalently by opioid use. Until WBE assessment is introduced to wastewater treatment processes, resources may be disproportionately dispersed throughout vulnerable communities.

### 6.3. Critical Factors to Consider during Implementation of WBE Approaches Using the NPDES

The effectiveness of this proposal and execution of the proposed plan is probable with minimal limitations, but there are some limitations that should be considered. Wastewater is a complex mixture that contains several contaminants that can alter the analysis of the wastewater sample. Wastewater can also be contaminated by rainwater and industrial discharges, and these additional contaminants could alter the results of the analysis. The analysis used to evaluate opioid metabolites should involve multiple steps that may be challenging to standardize and that require systematic controls [43]. Endo et al. 2020 has discussed an effective sampling strategy that has yielded significant results through the use of WBE approaches to assess opioid use [21]. To increase levels of precision and reduce uncertainty within the composite sampling process, a continuous sample should be utilized over a 24-h period, removal of small molecules should be filtered out using an Oasis HLB Solid-Phase Extraction (SPE) cartridge, and removal of large particles should be filtered out using a Durapore 0.22 μm filter [21]. In addition, to conduct an accurate data analysis when using the LC–MS or LC–MS/MS, internal calibration curves should be established [21]. Analyzing wastewater data could also bring forward privacy and ethical concerns. Even though it is difficult to link the wastewater data to a specific person, potential privacy problems and violation of civil liberties should be considered [44,45]. Affordability is also a key consideration when using WBE techniques because researchers have not determined the potential cost-savings associated with this process [8,43]. Further research is required to compare and quantify the analytical costs of WBE techniques to the costs that are associated with the logistics of opioid use and opioid overdoses to determine the true cost-effectiveness of the proposed process [8]. Such as the LC–MS or LC–MS/MS analysis, which is predominantly used for opioid detection in wastewater, is a more expensive, accurate equipment that is used in conjunction with WBE sampling processes [20,21]. Furthermore, to effectively execute the proposed solution, additional funding for personnel may be required to collect additional sampling and to conduct the additional analyses.

### 6.4. Prospective National Policy Consideration for the United States

Monitoring opioid metabolites by utilizing wastewater-based epidemiology approaches through the National Pollutant Discharge Elimination System (NPDES) has the potential to impact the nation as a new federal policy or regulation. The EPA enforces the NPDES under the Clean Water Act. This program grants the EPA the ability to propose an amendment to the CWA or to establish a new regulatory requirement under the NPDES regulatory program. The new policy would require each state to add opioid metabolites on their respective wastewater permits issued under the NPDES. With a policy of this caliber, the EPA would be able to collaborate with the CDC regarding wastewater-based epidemiology and opioid use within each state. This collaborative effort would allow the CDC to allocate more resources, raise awareness, and explore more solutions to address the national opioid epidemic.

## 7. Conclusions

We have discussed a potential solution that can assist public health officials with monitoring the ongoing opioid epidemic within metro areas through wastewater-based epidemiological approaches. A representative sample of opioids that are excreted through fecal matter can be analyzed through the collection of influent and effluent composite samples. Opioid metabolites analyzed in the wastewater should be placed in a database that is easily accessible by public health professionals and policy makers. After reviewing the results, there should be a focus on increasing awareness and allocating the proper resources to reduce opioid use within the affected area. Utilizing the public database could promote a collaboration between the EPA and the CDC to monitor opioid use through the NPDES regulatory permit program. In addition, even though a direct link to a specific person is difficult through wastewater analysis, protecting privacy and civil liberties should be considered.

Research has proven that wastewater-based epidemiology is an efficient, non-invasive tool that has yielded great success with monitoring opioid metabolites within the wastewater throughout various communities [11,20,21]. Moreover, areas within Europe, Canada, and Australia, have proven that nationalized wastewater monitoring programs are evolving analytical techniques that can monitor drug usage, and establish sufficient policies to address the opioid crisis within various communities [30,31,34]. When developing an innovative solution for an ongoing problem, such as the opioid epidemic in Metro Atlanta, further evaluation of the solution should be considered. Requiring WWTPs to monitor opioid discharges is feasible and has several potential social impacts that can benefit the people who are directly affected by opioid use. WBE approaches have been utilized by other nations to address their ongoing opioid epidemic.

The process of analysis discussed within this paper outlines the potential to broaden the policy and resources provided to communities throughout the U.S. that are battling opioid use, abuse, and overdoses. In addition, the mentioned approach within this paper to address the opioid epidemic in the U.S. would require a collaborative effort among the EPA and local public health officials to monitor opioid metabolites through the utilization of wastewater-based epidemiology on NPDES permits. As a result, this approach would allow the local public health officials to allocate more resources, raise awareness, and explore more solutions to address the emerging opioid epidemic within significantly impacted areas such as Metro Atlanta, Georgia. Furthermore, utilizing a national wastewater monitoring program to evaluate opioid use within affected areas is an approach that can be utilized globally within various countries.

## Figures and Tables

**Figure 1 ijerph-20-05282-f001:**
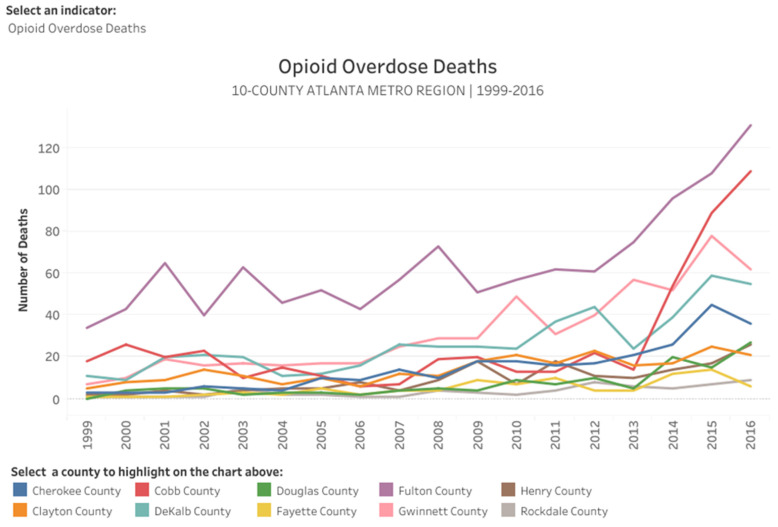
Opioid overdose deaths in Metro Atlanta from 1999 to 2016. Data adapted from the Atlanta Regional Commission (ARC) 2018 [4].

**Figure 2 ijerph-20-05282-f002:**
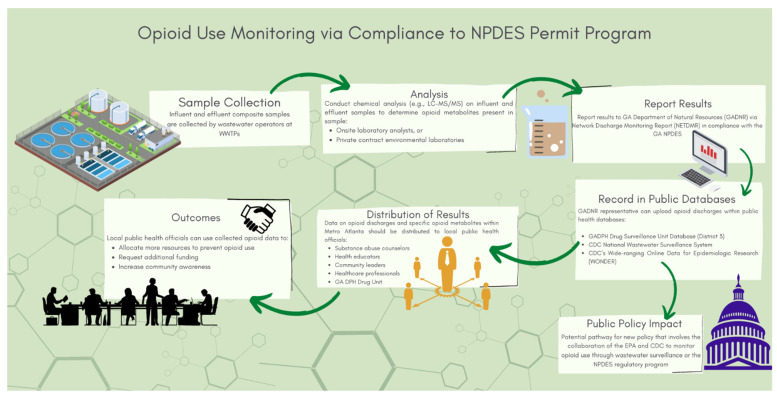
Streamlined approach on opioid use monitoring through compliance to the NPDES permit program.

## Data Availability

Publicly available datasets were analyzed in this study. The data can be found here: https://33n.atlantaregional.com/public-health/special-feature-opioid-overdose-deaths-rise-in-metro-region (accessed on 21 June 2022).

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
