# Peer review of "Utilizing a National Wastewater Monitoring Program to Address the U.S. Opioid Epidemic: A Focus on Metro Atlanta, Georgia"

_ijerph, 2023, doi:10.3390/ijerph20075282_

Round 1

Reviewer 1 Report

Review reports for the paper ijerph-2272572

This paper highlights serious problems. However, this work can be accepted after revising based on the following comments: 

1. Please explain more deeply figure 1. Why have the death toll of Cobb and Fulton Counties increased after 2013? What is the reason behind this? 

2. Why is the number of deaths in Fulton County higher than other counties? 

3. In the abstract authors claimed that this work was focusing on two issues: removing opioid contamination from wastewater and use of opioid residue data in wastewater as an alternative method for opioid exposure assessment in the community. However, no data and no methods for removing opioid contamination from wastewater were discussed in the research. 

4. Novelty of the research is not clear. The author should focus on the advantages and disadvantages of the existing methods, and why the author adopted it?

5. Please cite following water purification papers in your research: 

Abdiyev, K.Z., Maric, M., Orynbayev, B.Y. et al. Flocculating properties of 2-acrylamido-2-methyl-1-propane sulfonic acid-co-allylamine polyampholytic copolymers. Polym. Bull. 79, 10741–10756 (2022). https://doi.org/10.1007/s00289-021-03994-2.

Vassilis J. Inglezakis, Seitkhan Azat, Zhandos Tauanov, Sergey V. Mikhalovsky, Functionalization of biosourced silica and surface reactions with mercury in aqueous solutions, Chemical Engineering Journal, 2021, ISSN 1385-8947, https://doi.org/10.1016/j.cej.2021.129745

Author Response

Please see the attachment for a point-by-point response to reviewer's comments.

Reviewer 2 Report

The authors of the manuscript entitled "Utilizing a National Wastewater Monitoring Program to Address the U.S. Opioid Epidemic: A Focus on Metro Atlanta, Georgia" review the national and worldwide practice of wastewater monitoring for estimating levels of opioid usage and propose the framework for this monitoring at the official level within a local (Metro Atalanta) area.

The study addresses the acute and vital problem, which is related to raising public awareness, allocating funds and seeking solutions to address the ongoing opioid epidemic in the U.S. The work may be interesting for researchers and policymakers from other countries who face the similar challenges.

The authors propose incorporating the wastewater opioid monitoring within the National Pollutant Discharge Elimination System (NPDES) permit program. Yet, the study does not provide a detailed action plan as to how this integration can be accomplished. In particular, for me as a non-U.S. citizen, it is not clear which officials, and under which circumstances, would decide on the inclusion of opioid monitoring in the NPDES permits and which funds would cover the associated costs. It is worthwhile to mention that the LC-MS or LC-MS/MS analysis used for the opioid detection and quantitation in wastwater is perhaps even more expensive than all the other determinations put together within the GA NPDES permits.

In Section 5 the authors overview challenges related to wastewater-based opioid epidemiology. However, this account seems too short and superficial. I recommend the work [Subedi B, Burgard D (2019) Wastewater-Based Epidemiology: Estimation of Community Consumption of Drugs and Diets. In: Subedi B, Burgard DA, Loganathan BG (eds.) Wastewater-Based Epidemiology as a Complementary Approach to the Conventional Survey-Based Approach for the Estimation of Community Consumption of Drugs. ACS Symposium Series, Vol. 1319, pp.3-21. doi:10.1021/bk-2019-1319.ch001] for more details.

It is necessary to note that the list of opioids (L.244-247) to monitor is not at all exhaustive if monitoring opioid metabolites is concerned as well. For example, the major metabolite of hydrocodone is norhydrocodone [Zhou S (2016) Cytochrome P450 2D6: Structure, Function, Regulation and Polymorphism. CRC Press. pp. 164], and that of oxycodone is noroxycodone [Smith H, Passik S (2008) Pain and Chemical Dependency. Oxford University Press, USA. pp. 195], which have not been not mentioned by the authors. I suggest that the authors should thoroughly revise this list, taking into account main metabolites of the most popular opioids.

L.196-197: "opioid use was a major contributing factor for the increase of opioid overdoses throughout the nation" is too obvious a statement that should be reworded.

L.265-267: "Several municipalities are exploring solutions to reduce opioid discharges from their wastewater treatment process, but these processes are not universally used at WWTPs through the U.S." References are missing. What kind of solutions are in question?

L.277-278: "if opioid discharges are not monitored by the NPDES permit program, not only is marine health impacted, but these discharges could have an impact on water reuse processes and wastewater discharges." It is not clear how opioid discharges can have an impact on wastewater discharges.

Author Response

Please see the attachment for a point-by-point response to reviewer's comments. Thank you for your peer review.

Reviewer 3 Report

Manuscript Title: Utilizing a National Wastewater Monitoring Program to Address the U.S. Opioid Epidemic: A Focus on Metro Atlanta, Georgia 

Manuscript ID: ijerph-2272572

This article focuses on removing opioid contamination from wastewater and use of opioid residue data in wastewater as an alternative method for opioid exposure assessment in the community. The article proposes a new innovative approach for monitoring opioid use and analyzing trends by utilizing wastewater-based epidemiologic methods. The article is worthy of reading and is written elegantly. I have the following queries:

1.      Other than wastewater-based epidemiology, is there any other potential tools to present a snapshot of the overall health of a community based on what is being excreted in a pooled sample of sewage?

2.      Whether wastewater-based epidemiology can be used to monitor the spread and evolution of the SARS-CoV-2 virus?

3.      How opioids are treated and removed from the wastewater by using wastewater-based epidemiology? Please explain the mechanism.

4.      Could you mention the advantages of wastewater-based epidemiology over conventional surveillance methods like surveys, clinical testing, medical records, prescribing or sales data, human biomonitoring studies, crime statistics, etc.?

5.      What is the cost involved in applying wastewater-based epidemiology for different public health purposes?

6.      From the article it is understood that the wastewater-based epidemiology has typically been used as a complementary tool. Are there any examples of where wastewater-based epidemiology has been integrated across different public health domains?

7.      How do you address the high levels of uncertainty and lack of precision associated with the wastewater-based epidemiology?

8.      Enlist the limitations of wastewater-based epidemiology when compared with conventional surveillance methods mentioned above.

Thank You

Author Response

(The authors gave the same response as above.)

Round 2

Reviewer 1 Report

The paper was revised, thank you.